# Boosting Masked ECG-Text Auto-Encoders as Discriminative Learners

Hung Manh Pham [1]    Aaqib Saeed [2]    Dong Ma [1]

## Abstract

The accurate interpretation of Electrocardiogram (ECG) signals is pivotal for diagnosing cardiovascular diseases. Integrating ECG signals with accompanying textual reports further holds immense potential to enhance clinical diagnostics by combining physiological data and qualitative insights. However, this integration faces significant challenges due to inherent modality disparities and the scarcity of labeled data for robust cross-modal learning. To address these obstacles, we propose D-BETA , a novel framework that pre-trains ECG and Text data using a contrastive masked Auto-encoder architecture, uniquely combining generative and Boosted Discriminative capabilities for robust cross-modal representations. This is accomplished through masked modality modeling, specialized loss functions, and an improved negative sampling strategy tailored for cross-modal alignment. Extensive experiments on five public datasets across diverse downstream tasks demonstrate that D-BETA significantly outperforms existing methods, achieving an average AUC improvement of 15% in linear probing with only one percent of training data and 2% in zero-shot performance without requiring training data over state-of-the-art models. These results highlight the effectiveness of D-BETA , underscoring its potential to advance automated clinical diagnostics through multi-modal representations. [1]

## 1. Introduction

Electrocardiograms (ECGs), obtained through non-invasive electrode placement, provide a critical window into the heart's electrical activity by measuring voltage differences across specific anatomical regions. The standard 12-lead ECG, which captures unique electrical potential differences from each lead, plays a vital role in diagnosing a wide spectrum of cardiac conditions (e.g. arrhythmias). In recent years, significant progress has been made in leveraging deep learning techniques for automated ECG interpretation (Yan et al., 2019; Ebrahimi et al., 2020; Siontis et al., 2021). However, these supervised deep learning approaches often necessitate large volumes of expertly annotated data, which are frequently scarce and expensive to acquire. Self-supervised learning (SSL) has emerged as a compelling alternative, offering the potential to learn robust representations from abundant unlabeled ECG data. These learned representations can be effectively utilized for zero-shot learning on novel tasks and adapted via fine-tuning to specific downstream applications, thereby mitigating the reliance on extensive labeled datasets.

Numerous studies have explored the potential of SSL in the ECG domain, demonstrating its efficacy in learning representations from vast quantities of unlabeled data. These efforts generally fall into two main tracks: contrastive and generative approaches. Contrastive methods, exemplified by works such as (Chen et al., 2020; 2021; Chen & He, 2021; Grill et al., 2020; Kiyasseh et al., 2021; Oh et al., 2022; McKeen et al., 2024), aim to learn discriminative representations by maximizing the similarity between positive pairs (e.g., different augmentations of the same ECG signal) and minimizing the similarity between negative pairs (e.g., ECGs from different patients) within the embedding space. Conversely, generative approaches (Hu et al., 2023; Zhang et al., 2022a; 2023) focus on reconstructing the input data, typically by predicting masked or missing segments of the ECG signal, thereby learning to capture the underlying data distribution. Therefore, integrating both contrastive and generative approaches within a unified framework could leverage their complementary strengths, leading to a more powerful method for learning robust ECG-text representations (Kim et al., 2021; Li et al., 2022b; Song et al., 2024).

Despite advancements, existing ECG-based SSL approaches have largely overlooked the valuable information embedded within clinical text reports, which offer key insights into underlying cardiac conditions and have the potential to significantly enhance a model's diagnostic accuracy (Zhang et al., 2022c; Chen et al., 2022). This oversight highlights a

[1]Singapore Management University [2]Eindhoven University of Technology. Correspondence to: Dong Ma <dongma@smu.edu.sg>.

*Proceedings of the 42nd International Conference on Machine Learning*, Vancouver, Canada. PMLR 267, 2025. Copyright 2025 by the author(s).

[1]Our code and checkpoint are made available at `https://github.com/manhph2211/D-BETA`.

critical gap in the field: the lack of emphasis on jointly learning ECG-text cross-modal representations. Some recent efforts (Liu et al., 2024b; Lalam et al., 2023; Li et al., 2024; Liu et al., 2024a; Yu et al., 2024) have attempted to bridge this gap by integrating ECG signals and clinical reports through cross-modal contrastive learning, mainly employing relatively standard encoder models (e.g., ResNet (He et al., 2016), Bert (Devlin et al., 2019)). This makes the potential of learning robust representations that capture the intricate interplay between ECG signals and their corresponding textual descriptions, shown in generative approaches remain largely unexplored. Moreover, the prevailing reliance on these contrastive methods presents inherent limitations. They depend on the availability of negative samples and often struggle to capture cross-modal relationships effectively due to difficulties in defining appropriate negative pairings across different modalities.

In this work, we depart from relying solely on contrastive learning or stand-alone generative approaches for cross-modal representation learning. We introduce D-BETA , a novel hybrid framework that synergistically integrates both learning paradigms to effectively capture fine-grained input details and discriminative ECG-text features. Our approach employs a transformer-based encoder specifically for ECG signals and a well-pre-trained language model for clinical text encoder in a masked auto-encoder architecture, together with tailored loss functions that promote the joint learning of robust cross-modal representations. Additionally, we introduce a nearest-neighbor negative sampling strategy, a crucial refinement often overlooked in previous methods, to ensure that negative samples are contextually selected and thereby, enhance the discriminative capability of the learned representations. To rigorously evaluate the efficacy of D-BETA , we conduct extensive experiments on various public ECG datasets and demonstrate that our method significantly outperforms recent state-of-the-art baselines across all evaluation settings.

## 2. Related Work

**ECG Self-supervised Learning.** Self-supervised learning (SSL) has been shown to work effectively across various modalities, including vision (Li et al., 2022a; Han et al., 2021), language (Devlin et al., 2019; He et al., 2020; Chung et al., 2024), and time-series data (Tonekaboni et al., 2021; Zhang et al., 2022b; Saeed et al., 2019). Particularly, recent advances in applying SSL to ECG signals have demonstrated that models can learn meaningful representations from large amounts of unlabeled data, which is crucial in medical domains where labeled datasets are often limited and expensive to acquire. Here, we discuss two common SSL approaches: generative and contrastive, which have seen notable progress in ECG representation learning in recent years.

Early contrastive methods such as SimCLR (Chen et al., 2020), MoCo (Chen et al., 2021), SimSiam (Chen & He, 2021), and BYOL (Grill et al., 2020) introduced the concept of maximizing agreement between augmented views of the same data sample by employing augmentation strategies to create challenging positive and negative pairs. In the context of ECG signals, recent approaches like 3KG (Gopal et al., 2021) apply physiologically inspired spatial and temporal augmentations, using vectorcardiogram (VCG) transformations to capture the three-dimensional spatiotemporal characteristics of the heart's electrical activity. Similarly, CLOCS (Kiyasseh et al., 2021) developed Contrastive Multi-Segment Coding (CMSC), which enhances the model's ability to handle varying ECG signal characteristics across different axes-space, time, and patients. Building on this, (Oh et al., 2022) incorporates Wav2vec 2.0 (Baevski et al., 2020), CMSC, and random lead masking to simulate different global and local lead configurations during training, thereby improving model robustness and achieving impressive results on ECG downstream tasks.

On the other hand, generative approaches (Hu et al., 2023; Zhang et al., 2022a; Na et al., 2024) are less prevalent, but play a crucial role in ECG SSL. These methods focus on capturing the underlying structure of the data by training auto-encoder models to generate or reconstruct masked input data, enabling the model to understand and represent key features and patterns. For instance, ST-MEM (Na et al., 2024) utilizes a masked auto-encoder with a spatio-temporal patchifying technique to model relationships in 12-lead ECG signals. Additionally, the Cross-Reconstruction Transformer (CRT) (Zhang et al., 2023) employs frequency-domain and temporal masking to reconstruct missing ECG segments, demonstrating the innovative use of generative SSL in ECG analysis.

**ECG-Text Multi-modal Representation Learning.** Multi-modal representation learning combines information from different data types, shown to effectively improve model performance (Lin et al., 2024; Du et al., 2023). Particularly, pioneering works like CLIP-based models (Radford et al., 2021; Rasheed et al., 2023; Zhai et al., 2023) have proven the power of contrastive learning in aligning visual and textual modalities, achieving strong generalizations across a broad range of tasks. Applying similar ideas to the ECG domain, recent efforts (Lalam et al., 2023; Yu et al., 2024; Liu et al., 2024a;b) show promising progress in the field. Among them, MERL (Liu et al., 2024b) leverages cross-modal and uni-modal alignment techniques together with test-time clinical knowledge enhancement, which notably generalizes ECG and text-based medical zero-shot classification tasks. However, they often utilize genetic architectures (e.g., ResNet ECG encoder, Bert-based

text encoder) and especially overlook the critical role of negative sample selection for contrastive learning and lacks exploring generative approaches for fine-grained multi-modal learning, limiting performance in end tasks.

## 3. Method

We propose D-BETA , a framework designed to learn generalizable cross-modal representations by aligning ECG signals and corresponding medical text reports. D-BETA leverages masked language modeling (MLM) and masked ECG modeling (MEM) to reconstruct randomly masked segments within the input text and ECG signals, respectively. This encourages the model to learn fine-grained features within each modality. Furthermore, we introduce ETS (ECG-Text Sigmoid) loss, as inspired by SigLIP (Zhai et al., 2023), and a nearest-neighbor negative sampling strategy. These directly promote discriminative representation learning and enhance cross-modal alignment, besides the ECG-text matching (ETM) learning task.

Figure 1 depicts the overall architecture of D-BETA , which comprises two main branches. The ECG encoder utilizes a transformer-based architecture (Vaswani et al., 2023) to process the input ECG signals and generate corresponding representations, denoted as $\mathbf{H}_x \in \mathbb{R}^{L_x \times d}$, where $L_x$ represents the sequence length of the ECG signal and $d$ represents the embedding dimension. The text encoder utilizes the recent pre-trained Flan-T5 model (Chung et al., 2024) which, to our knowledge, has not been previously applied to this ECG domain, to extract high-level semantic embeddings from the clinical text, denoted as $\mathbf{H}_t \in \mathbb{R}^{L_t \times d}$, where $L_t$ represents the sequence length of the text. These encoder outputs are then passed through a fusion module, which employs a cross-attention mechanism to integrate information from both modalities, generating fused representations denoted as $\mathbf{H}_f \in \mathbf{R}^{(L_x+L_t) \times d}$. The model subsequently employs three distinct heads: two decoders, responsible for reconstructing the masked portions of the ECG signal ($\hat{X}$) and text ($T_m$), respectively, and a contrastive prediction head for ECG-text matching. Additionally, we introduce two projection heads, $g_x$ and $g_t$, following the ECG and text encoders, respectively. These projection heads, along with the ETS loss, facilitate learning discriminative representation between these modalities. The model is trained by jointly optimizing four loss functions: masked language modeling loss ($\mathcal{L}_{MLM}$), masked ECG modeling loss ($\mathcal{L}_{MEM}$), ECG-text matching loss ($\mathcal{L}_{ETM}$), and the ETS loss ($\mathcal{L}_{ETS}$ ). The subsequent subsections provide a detailed description of each component within the D-BETA framework.

### 3.1. Multi-Modal Masked Auto-Encoders.

**ECG Encoder.** We implement the ECG encoder (denoted as $\mathcal{F}_x$) based on a transformer architecture, which was orig-

inally developed for efficiently processing sequential data in parallel (Vaswani et al., 2023). We first follow (Oh et al., 2022) to apply a masking strategy to the ECG input $\mathbf{X} \in \mathbb{R}^{L \times C}$ to encourage robust feature learning, where $L$ is the length of the signal and $C$ is the number of channels. Specifically, we leverage random lead masking as an on-the-fly augmentation where each lead randomly masked with a probability of $p = 0.5$ during pre-training. Furthermore, we use a dropout layer on the input with $p = 0.1$ to enable masking modeling. We then pass the masked input into a series of convolutional layers, each followed by GELU activation functions and group normalization. The extracted features are subsequently projected into a 768-dimensional space. Following that, we add a convolutional positional encoding layer to preserve the temporal order of the ECG sequence. Next, we employ eight transformer encoder layers, each including a multi-head self-attention mechanism that allows the model to attend to different parts of the input sequence simultaneously. We conduct an experiment exploring the effects of different numbers of transformer layers in Section 4.3.

**Text Encoder.** For our text encoder, we utilize the Flan-T5-base encoder (denoted as $\mathcal{F}_t$), which outputs 768-dimensional embeddings. The input to the encoder consists of token indices generated by the Flan-T5 tokenizer, represented as $\mathbf{T} \in \mathbb{Z}^M$, where $M$ is the maximum sequence length. Flan-T5 is an advanced version of the T5 model (Raffel et al., 2023), which has been pre-trained on a massive and diverse text dataset covering numerous tasks, such as summarization and question answering. Note that our text encoder is fine-tuned during the pre-training stage. In our attempt to demonstrate the effectiveness of recent Flan-T5 in the ECG-domain-based context, we also conduct an ablation with various text encoders in Section 4.3.

**Fusion Module.** The fusion module begins with linear projections that map the outputs of the ECG and language encoders to a 768-dimensional space. We apply modality-specific embeddings to the projected features to distinguish between ECG and text data. Importantly, we employ cross-attention to integrate the ECG and textual information, allowing each modality to inform the other by learning the relevant features. This cross-attention mechanism is crucial as it enables the model to leverage the complementary strengths of both ECG and text data more effectively.

**Decoders and Self-Supervised Tasks.** After the fusion module, three network heads are introduced, each associated with a specific task or loss function: masked language modeling (MLM), masked ECG modeling (MEM), and ECG-text matching (ETM). MLM and MEM are designed for reconstruction tasks, while ETM adopts a contrastive learning approach to align the different modalities. We detail each task and its corresponding loss function below:

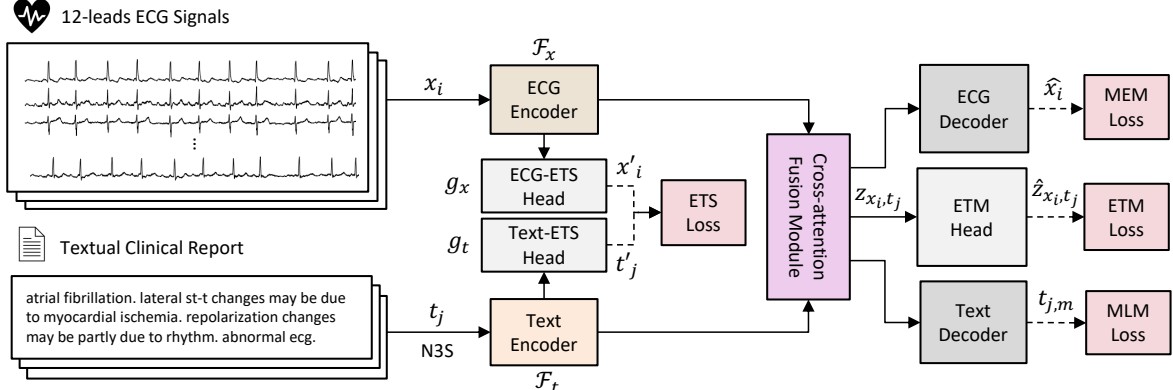

*Figure 1.* Illustration of our contrastive masked ECG-language modeling technique.

*Masked Language Modeling (MLM).* The MLM task consists of a dense layer that outputs a probability distribution over the vocabulary. The MLM head focuses on predicting the masked tokens in the input text sequence, encouraging the model to learn contextualized word embeddings through a reconstruction task. We use the cross-entropy (CE) loss for MLM, as shown in Equation 1:

$$\mathcal{L}_{\text{MLM}} = -\frac{1}{\mathcal{B}} \sum_{j=1}^{\mathcal{B}} \sum_{m \in \mathcal{M}_j} \log P(t_{j,m}|\mathbf{t}_{j \setminus \mathcal{M}_j}; \theta), \quad (1)$$

where $\mathcal{B}$ represents the batch size, $\mathcal{M}_j$ is the set of masked positions in the $j^{th}$ sequence, $t_{j,m}$ is the masked token at position $m$ in the $j^{th}$ sequence, $\mathbf{t}_{j \setminus \mathcal{M}_j}$ represents the $j^{th}$ input sequence with masked tokens removed, and $\theta$ represents the model parameters.

*Masked ECG Modeling (MEM).* Similar to MLM, the MEM task aims to reconstruct the masked ECG inputs. It consists of a linear embedding layer that maps the input sequence to a lower-dimensional space (384), followed by learnable mask tokens that represent the missing portions of the sequence. We apply positional encoding to preserve the temporal structure of the ECG data. Subsequently, we use a multi-layer transformer decoder to model the dependencies within the sequence. Finally, a linear projection layer produces the predicted ECG signal ($\hat{\mathbf{x}}_i$). We train the MEM head using the mean squared error (MSE) between this predicted signal and the ground truth signal ($\mathbf{x}_i$), as shown in Equation 2:

$$\mathcal{L}_{\text{MEM}} = \frac{1}{\mathcal{B}} \sum_{i=1}^{\mathcal{B}} ||\hat{\mathbf{x}}_i - \mathbf{x}_i||_2^2 \quad (2)$$

*ECG-Text Matching (ETM).* Finally, we use ETM to promote alignment between ECG signals and their corresponding text reports, which further supports the fused feature space learning, together with generative aspects from MLM

and MEM. This is formulated as a binary classification task, where the ETM task's head consists of a single dense layer that outputs a scalar $\hat{z}_{\mathbf{x}_k, \mathbf{t}_k}$ representing the predicted probability. The ETM loss is defined as the binary cross-entropy loss:

$$\mathcal{L}_{\text{ETM}} = -\frac{1}{\mathcal{B}} \sum_{k=1}^{\mathcal{B}} \Big[ y_k \log \sigma(\hat{z}_{\mathbf{x}_k, \mathbf{t}_k})$$
$$+ (1 - y_k) \log(1 - \sigma(\hat{z}_{\mathbf{x}_k, \mathbf{t}_k})) \Big], \quad (3)$$

where $\sigma$ is the sigmoid function, $y_k = 1$ if $(\mathbf{x}_k, \mathbf{t}_k)$ is a positive pair, and $y_k = 0$ otherwise.

### 3.2. ECG-Text Discriminative Learners.

**ETS Loss Function.** In multi-modal masked auto-encoder architectures such as (Chen et al., 2022), contrastive learning's effectiveness can be limited by the inherent tension between the reconstruction-focused generative tasks of autoencoders and the discriminative nature of contrastive learning. They are more biased for learning to reconstruct masked inputs in generative manners. This can hinder the model's capability to learn discriminative features useful for downstream tasks, such as zero-shot inference or linear probing. Furthermore, although the ETM loss in such architectures can serve as a form of contrastive loss, it may not be sufficient for building a robust ECG encoder. Specifically, the ETM module is primarily designed for binary classification based on fused features rather than directly enhancing the discriminative power of individual encoders. This limitation can restrict the model's ability to produce high-quality multimodal embeddings.

Therefore, we propose strengthening discriminative aspect in multi-modal masked auto-encoder architectures using ETS loss, as shown in formula 4. This approach avoids the costly global normalization of softmax-based contrastive

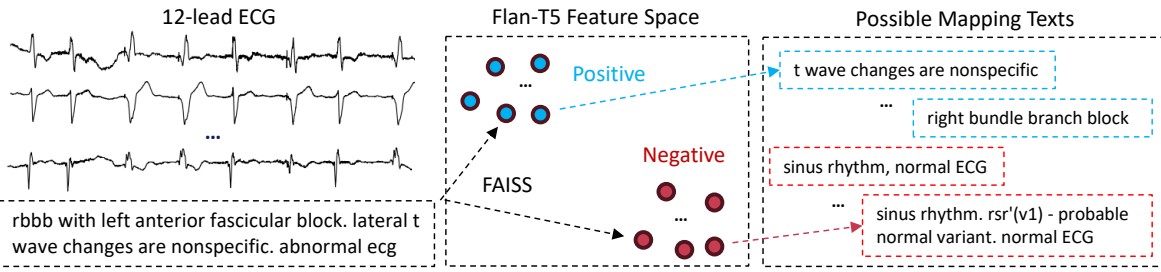

Figure 2. Illustration of N3S in selecting negative samples within Flan-T5 space.

losses by operating independently on each ECG-text pair (sigmoid-based), improving memory efficiency and scalability. We introduce two additional network heads to the ECG and text encoders, respectively. Each head consists of a pooling layer, a Tanh activation function, and a dense layer, enabling them to output 768-dimensional embeddings (denoted as $\mathbf{x}'_i \in \mathbb{R}^{768}$ for the $i^{th}$ ECG sample and $\mathbf{t}'_j \in \mathbb{R}^{768}$ for the $j^{th}$ text report).

$$\mathcal{L}_{\text{ETS}} = -\frac{1}{\mathcal{B}} \sum_{i=1}^{\mathcal{B}} \sum_{j=1}^{\mathcal{B}} \log\left(\frac{1}{1 + e^{-y_{ij}\mathbf{x}'^{\top}_i \mathbf{t}'_j}}\right), \quad (4)$$

where $y_{ij} = 1$ for positive (matching) ECG-text pairs, and $y_{ij} = -1$ otherwise.

**Nearest-neighbor-based Negative Sampling (N3S).** In contrastive learning, the selection of negative samples significantly impacts the training process (Xu et al., 2022). Conventional methods often employ random sampling, where negative text reports are chosen randomly to replace positive texts. However, this approach may lead to false negative selection, especially in medical datasets, where randomly chosen reports might share substantial similarities with the positive reports, hindering effective contrastive learning. This is discussed further in the *Appendix* A.2.

Therefore, we propose nearest-neighbor negative sampling (N3S), which selects negative samples based on their dissimilarity in the Flan-T5's feature space (Figure 2), ensuring they are sufficiently distinct from the positive samples while remaining semantically related to the domain. Specifically, we first utilize pre-trained Flan-T5 (small) to generate vector representations, denoted as $\mathbf{v}_t \in \mathbb{R}^{512}$, for each text report $t$ in the training dataset $\mathcal{D}_{train}$. These embeddings capture the semantic meaning of the reports. During training, for a given ECG and its corresponding positive text report $(x_k, t_k^+)$ in half of the training batch $\mathcal{B}$, the negative report $t_k^-$ is selected as one of the top 64 largest cosine distance reports from the positive report's embedding $\mathbf{v}_{t_k^+}$. As the training progresses with batches being updated randomly, the negative samples continually change, introducing variability while maintaining domain relevance.

To perform this process, we employ FAISS (Facebook AI Similarity Search) (Douze et al., 2024), a high-performance library designed for indexing and searching large collections of dense vectors. FAISS allows us to apply N3S to large-scale datasets in a computationally tractable manner.

## 4. Experiments

### 4.1. Implementation Details.

#### 4.1.1. PRE-TRAINING TASK.

**Pre-train Dataset.** In the pre-training stage, we utilize the MIMIC-IV-ECG v1.0 database (Gow et al., 2023), which includes 800,035 paired samples derived from 161,352 unique subjects. This dataset contains numerous 10-second ECG recordings sampled at 500 Hz and the corresponding text reports. Each ECG recording will have several reports, and we simply merge them into one single report (diagnosis). We apply some necessary processing steps to prepare the custom dataset for training (e.g., remove empty or containing NaN ECG recordings and clean text by using lowercase, strip, and punctuation removal), which eventually yields a training size of 779891 samples. We provide representative examples of ECG-text pairs in *Appendix* A.1.

**Experimental Configurations.** For model training, we use the Adam optimizer with a learning rate of $5 \times 10^{-5}$ and use a tri-stage scheduler with ratios of 0.1, 0.4, and 0.5 for learning rate adjustments. The optimizer is configured with $\beta_1 = 0.9$, $\beta_2 = 0.98$, an epsilon value of $1 \times 10^{-6}$, and a weight decay of 0.01. We pre-train the proposed model for 300000 steps, maintaining a batch size of 128. The quantitative experiments are conducted on a single NVIDIA H100-80GB GPU.

#### 4.1.2. DOWNSTREAM TASKS.

**Downstream Datasets.** We evaluate our pre-trained encoders on five widely-used public datasets: PhysioNet 2021 (Reyna et al., 2021), PTB-XL (Wagner et al., 2020), CSN (Zheng et al., 2022), CPSC2018 (Liu et al., 2018), and CODE-test (Ribeiro et al., 2020). We summarize the key information of each dataset as follows:

*Table 1.* Performance for 5 lead combinations in diagnosis classification (Dx., by CinC scores scaled by 100) and patient identification (Id., by %). P-N-lead indicates N zero-padded unavailable leads.

| Methods | Tasks | # Leads | | | | |
|---|---|---|---|---|---|---|
| | | 12-lead | P-6-lead | P-3-lead | P-2-lead | P-1-lead |
| W2V (Baevski et al., 2020) | Dx. | 71.4 | 64.3 | 67.6 | 61.1 | 52.5 |
| | Id. | 49.2 | 41.1 | 47.0 | 41.4 | 24.7 |
| CMSC (Kiyasseh et al., 2021) | Dx. | 62.5 | 52.2 | 57.5 | 50.7 | 40.6 |
| | Id. | 51.3 | 39.2 | 51.0 | 37.8 | 22.7 |
| 3KG (Gopal et al., 2021) | Dx. | 60.0 | 51.5 | 56.3 | 50.5 | 41.8 |
| | Id. | 40.7 | 32.0 | 36.7 | 31.0 | 19.8 |
| SimCLR(RLM) (Chen et al., 2020) | Dx. | 57.8 | 49.7 | 53.5 | 48.4 | 39.3 |
| | Id. | 35.3 | 28.9 | 36.8 | 30.4 | 19.2 |
| W2V+CMSC (Oh et al., 2022) | Dx. | 71.7 | 61.6 | 65.6 | 58.6 | 48.2 |
| | Id. | 55.0 | 43.7 | 46.6 | 41.0 | 28.0 |
| W2V+CMSC+RLM (Oh et al., 2022) | Dx. | 73.2 | 66.2 | 71.4 | 65.6 | 55.4 |
| | Id. | 57.7 | 45.9 | 54.8 | 45.7 | 31.3 |
| **D-BETA** | **Dx.** | **85.7** | **81.1** | **84.2** | **81.9** | **76.5** |
| | **Id.** | **65.4** | **57.3** | **60.5** | **57.7** | **41.1** |

*Table 2.* Performance comparison (AUC in %) across multiple methods and datasets. The results are shown for different percentages of training data used (1%, 10%, 100%).

| Methods | PTBXL-Super | | | PTBXL-Sub | | | PTBXL-Form | | | PTBXL-Rhythm | | | CPSC2018 | | | CSN | | |
|---|---|---|---|---|---|---|---|---|---|---|---|---|---|---|---|---|---|---|
| | 1% | 10% | 100% | 1% | 10% | 100% | 1% | 10% | 100% | 1% | 10% | 100% | 1% | 10% | 100% | 1% | 10% | 100% |
| SimCLR (Chen et al., 2020) | 63.41 | 69.77 | 73.53 | 60.84 | 68.27 | 73.39 | 54.98 | 56.97 | 62.52 | 51.41 | 69.44 | 77.73 | 59.78 | 68.52 | 76.54 | 59.02 | 67.26 | 73.20 |
| BYOL (Grill et al., 2020) | 71.70 | 73.83 | 76.45 | 57.16 | 67.44 | 71.64 | 48.73 | 61.63 | 70.82 | 41.99 | 74.40 | 77.17 | 60.88 | 74.42 | 78.75 | 54.20 | 71.92 | 74.69 |
| BarlowTwins (Zbontar et al., 2021) | 72.87 | 75.96 | 78.41 | 62.57 | 70.84 | 74.34 | 52.12 | 60.39 | 66.14 | 50.12 | 73.54 | 77.62 | 55.12 | 72.75 | 78.39 | 60.72 | 71.64 | 77.43 |
| MoCo-v3 (Chen et al., 2021) | 73.19 | 76.65 | 78.26 | 55.88 | 69.21 | 76.69 | 50.32 | 63.71 | 71.31 | 51.38 | 71.66 | 74.33 | 62.13 | 76.74 | 75.29 | 54.61 | 74.26 | 77.68 |
| SimSiam (Chen & He, 2021) | 73.15 | 72.70 | 75.63 | 62.52 | 69.31 | 76.38 | 55.16 | 62.91 | 71.31 | 49.30 | 69.47 | 75.92 | 58.35 | 72.89 | 75.31 | 58.25 | 68.61 | 77.41 |
| TS-TCC (Eldele et al., 2021) | 70.73 | 75.88 | 78.91 | 53.54 | 66.98 | 77.87 | 48.04 | 61.79 | 71.18 | 43.34 | 69.48 | 78.23 | 57.07 | 73.62 | 78.72 | 55.26 | 68.48 | 76.79 |
| CLOCS (Kiyasseh et al., 2021) | 68.94 | 73.36 | 76.31 | 57.94 | 72.55 | 76.24 | 51.97 | 57.79 | 72.65 | 47.19 | 71.88 | 76.31 | 59.59 | 77.78 | 77.49 | 54.38 | 71.93 | 76.13 |
| ASTCL (Wang et al., 2023) | 72.51 | 77.31 | 81.02 | 61.86 | 68.77 | 76.51 | 44.14 | 60.93 | 66.99 | 52.38 | 71.98 | 76.05 | 57.90 | 77.01 | 79.51 | 56.40 | 70.87 | 75.79 |
| CRT (Zhang et al., 2023) | 69.68 | 78.24 | 77.24 | 61.98 | 70.82 | 78.67 | 46.41 | 59.49 | 68.73 | 47.44 | 73.52 | 74.41 | 58.01 | 76.43 | 82.03 | 56.21 | 73.70 | 78.80 |
| ST-MEM (Na et al., 2024) | 61.12 | 66.87 | 71.36 | 54.12 | 57.86 | 63.59 | 55.71 | 59.99 | 66.07 | 51.12 | 65.44 | 74.85 | 56.69 | 63.32 | 70.39 | 59.77 | 66.87 | 71.36 |
| MERL (Liu et al., 2024b) | 82.39 | 86.27 | 88.67 | 64.90 | 80.56 | 84.72 | 58.26 | 72.43 | 79.65 | 53.33 | 82.88 | 88.34 | 70.33 | 85.32 | 90.57 | 66.60 | 82.74 | 87.95 |
| **D-BETA** | **83.15** | **88.36** | **90.11** | **77.74** | **82.92** | **85.15** | **70.10** | **78.91** | **83.98** | **86.61** | **92.83** | **96.71** | **85.46** | **91.35** | **94.92** | **80.04** | **87.36** | **90.71** |

*PhysioNet 2021.* This contains ECG samples (500 Hz) ranging between 5 and 144 seconds. We process and fine-tune the subsets as described in (Oh et al., 2022) to validate the pre-trained ECG encoder in two downstream tasks: 1) 26-multi-label cardiac arrhythmia classification (Dx.); 2) patient identification (Id.), predicting patient ownership of ECG recordings.

*PTB-XL.* The PTB-XL dataset includes 21,837 ECG signals collected from 18,885 patients. Each sample has a 12-lead ECG recording sampled at 500 Hz over 10 seconds and corresponding cardiac labels. We follow (Liu et al., 2024b) to split this dataset, including four sub-groups (super, sub, form, and rhythm). We consider them as the four separated datasets and prepare each of them with the same train, val, and test set as in the original paper (Wagner et al., 2020).

*CSN.* This dataset consists of 23,026 ECG recordings sampled at 500 Hz for 10 seconds with 38 distinct labels, which also supports the evaluation in a classification task. We use 70%:10%:20% data split as processed in (Liu et al., 2024b).

*CPSC2018.* The dataset contains 6,877 standard 12-lead ECG recordings (500 Hz), which cover 9 distinct categories. Similarly, we also use the same data configuration following (Liu et al., 2024b).

*CODE-test*: This contains 827 12-lead ECG samples (400 Hz) at varying lengths covering 6 abnormalities, annotated by experienced residents and medical students. We resample the signals to 500 Hz and adjust the lengths to 10 seconds. We provide more detail about this dataset in *Appendix* A.1.

**Experimental Configurations.** To evaluate our model's performance on downstream tasks, we conduct three experiments: 1) First, integrating a linear layer on top of the pre-trained ECG encoder and fine-tuning the entire model to test its efficacy in two tasks within the Physionet 2021 dataset: Dx. (by CinC score) and Id. (by % accuracy). We report the results with five cases of lead combinations, as presented in (Oh et al., 2022); 2) Second, we also implement a linear classifier but keep the ECG encoder frozen. This linear probing approach is applied at different training set

*Table 3.* Zero-shot performance (AUC in %) comparison across multiple datasets.

| Methods | PTBXL-Super | PTBXL-Sub | PTBXL-Form | PTBXL-Rhythm | CPSC2018 | CSN | Average |
|---------|-------------|-----------|------------|--------------|----------|-----|---------|
| MERL | 74.2 | 75.7 | 65.9 | 78.5 | **82.8** | 74.4 | 75.3 |
| **D-BETA** | **76.2** | **75.9** | **66.1** | **88.6** | 80.1 | **76.3** | **77.1** |

*Table 4.* Zero-shot performance (AUC in %) under data distribution shift.

| Source Domain | | | PTBXL-Super | | CPSC2018 | | CSN | |
|---------------|--|--|-------------|--|----------|--|-----|--|
| **Target Domain** | Zero-shot | Training Ratio | CPSC2018 | CSN | PTBXL-Super | CSN | PTBXL-Super | CPSC2018 |
| SimCLR (Chen et al., 2020) | ✗ | 100% | 69.62 | 73.05 | 56.65 | 66.36 | 59.74 | 62.11 |
| BYOL (Grill et al., 2020) | ✗ | 100% | 70.27 | 74.01 | 57.32 | 67.56 | 60.39 | 63.24 |
| BarlowTwins (Zbontar et al., 2021) | ✗ | 100% | 68.98 | 72.85 | 55.97 | 65.89 | 58.76 | 61.35 |
| MoCo-v3 (Chen et al., 2021) | ✗ | 100% | 69.41 | 73.29 | 56.54 | 66.12 | 59.82 | 62.07 |
| SimSiam (Chen & He, 2021) | ✗ | 100% | 70.06 | 73.92 | 57.21 | 67.48 | 60.23 | 63.09 |
| TS-TCC (Eldele et al., 2021) | ✗ | 100% | 71.32 | 75.16 | 58.47 | 68.34 | 61.55 | 64.48 |
| CLOCS (Kiyasseh et al., 2021) | ✗ | 100% | 68.79 | 72.64 | 55.86 | 65.73 | 58.69 | 61.27 |
| ASTCL (Wang et al., 2023) | ✗ | 100% | 69.23 | 73.18 | 56.61 | 66.27 | 59.74 | 62.12 |
| CRT (Zhang et al., 2023) | ✗ | 100% | 70.15 | 74.08 | 57.39 | 67.62 | 60.48 | 63.33 |
| ST-MEM (Na et al., 2024) | ✗ | 100% | 76.12 | 84.50 | 62.27 | 75.19 | 73.05 | 64.66 |
| MERL (Liu et al., 2024b) | ✓ | 0% | **88.21** | 78.01 | 76.77 | 76.56 | 74.15 | **82.86** |
| **D-BETA** | ✓ | **0%** | 72.09 | **79.11** | **77.12** | **82.91** | **76.24** | 80.10 |

*Table 5.* ECG interpretation comparison (AUC in %): Human experts vs. DNN (Ribeiro et al., 2020) vs. D-BETA .

| Cardio Resident | Emergency Resident | Medical Student | DNN | D-BETA (Zero-shot) |
|-----------------|--------------------|-----------------|-----|---------------------|
| 92.07 | 90.52 | 93.61 | 96.59 | **96.79** |

sizes (1%, 10%, and 100%) to assess the macro AUC score (%) on the PTB-XL, CSN, and CPSC2018 test datasets, facilitating a comparison with our baseline (Liu et al., 2024b); 3) Finally, we investigate zero-shot classification (AUC) on PTB-XL, CSN, CPSC2018 and CODE-test datasets. Here, the texts used are obtained by passing the category names into GPT-4o to capture better medical context. The detailed configuration on each experiment is mentioned in *Appendix* A.1.

## 4.2. Experimental Results.

**Full Fine-Tuning Evaluation.** As shown in Table 1, our method consistently outperforms previous approaches (Oh et al., 2022) in both examined tasks. In the classification task, our model achieves 85.7% accuracy with all 12 leads, significantly higher than the best baseline (W2V+CMSC+RLM), which is 73.2%. This number is even lower than our setting with only 1 lead usage (76.5%). Interestingly, the 3-lead combination yields the second-highest result, only 1.5% lower than using all leads, while the 2-lead and 6-lead combinations produce comparable results, both around 81.5%. This suggests that the selected leads (I, II, V2) capture sufficient information for accurate performance. A similar pattern emerges in the identification task, where our model achieves 41.1% accuracy with a single lead, 60.5% with 3 leads, and 65.4% with 12 leads, surpassing the best baseline by 7%.

**Linear Probing Evaluation.** Table 2 presents the linear probing results, where our method demonstrates a clear advantage over the baseline approaches. Notably, with only 1% of the training data, our method shows a substantial improvement over MERL, especially in CSN (14% enhancement) and PTBXL-Rhythm (33%) datasets. Similarly, impressive results are observed at 10% and 100% of the data. For example, on the PTBXL-Rhythm dataset, our method achieves approximately a 10% improvement at the 10% configuration. On the CPSC2018 dataset, we also observe a considerable increase from 90.57% to 94.92% when using 100% of the training data.

**Zero-shot Evaluation.** We first compare our results and the best results of MERL in zero-shot settings across six datasets, as shown in Table 3. On average, our method achieves 77%, outperforming MERL by 2%. Notably, MERL performs impressively on the CPSC2018 dataset, while its results on the other five datasets are consistently lower than ours. Next, we extend the comparison of our method with MERL and other SSL baselines (Liu et al., 2024b) under data distribution shifts. Specifically, we compare linear probing (100% training size) of SSL methods with MERL's and our zero-shot approach. In this setup, the source domain and target domain share some common categories. Details on this implementation can be found in *Appendix* A.1. As shown in Table 4, our results surpass MERL and other SSL methods, except when CPSC2018 is the target domain, which aligns with our previous obser-

vations. Finally, Table 5 shows that our zero-shot model outperforms three experienced cardiologists (over 3%) and also the in-domain model (Ribeiro et al., 2020), i.e., trained with millions of annotated ECG examples [2]. We discuss more on zero-shot settings in *Appendix* A.3.

To better understand how our method improves downstream performance, we visualize and compare the t-SNE embeddings generated by our ECG encoder on the CSN test set with those from MERL. For clearer visualization, we include only samples from unique categories and exclude categories with fewer than 50 samples. Figure 3 reveals that our embeddings show more well-defined and distinct clusters representing different ECG diagnoses.

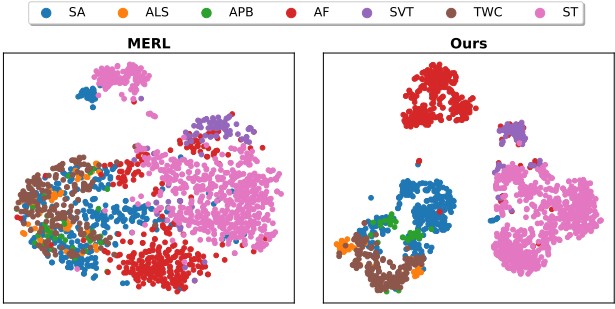

*Figure 3.* T-SNE visualization on the CSN test set.

### 4.3. Ablation Studies.

We evaluate the impact of the key model components, the choice of language encoders, and varying the number of transformer layers in the ECG encoder for ablation studies. Here, we focus on three downstream tasks, including full fine-tuned diagnosis classification (results across five lead combinations), linear probing at 1% training size, and zero-shot classification (results across PTB-XL, CSN, and CPSC2018 datasets) using category name [3].

*Table 6.* Effects of the model components: ① Flan-T5, ② ETS Loss, ③ N3S.

| ① | ② | ③ | Full fine-tune | Linear probing | Zero-shot |
|---|---|---|---|---|---|
| ✓ | ✓ | ✓ | **81.88 ± 3.52** | **80.52 ± 6.08** | **72.50 ± 9.01** |
| ✓ | ✓ | | 80.93 ± 3.74 | 78.29 ± 6.19 | 70.61 ± 8.10 |
| ✓ | | | 78.29 ± 3.87 | 67.19 ± 6.14 | – |
| | | | 76.81 ± 3.96 | 63.50 ± 6.95 | – |

**Effects of Key Components in D-BETA .** We systematically remove one component at a time from the default proposed model to assess the contribution of different model

---

[2]Medical students outperform residents due to recent frequently focused training, aligning with prior work (Ribeiro et al., 2020).

[3]We do not use GPT-4o for context enhancement, as our objective is to provide the core impact of our proposed components in the context of ablation studies.

components, including Flan-T5, ETS , and N3S. Specifically, we start by eliminating the N3S and train the model with randomly selected negative samples. Subsequently, we take the ETS loss away to assess its effectiveness in capturing rich representative embeddings in both encoders. Lastly, by replacing the Flan-T5 language encoder with a standard Bert-base architecture (Devlin et al., 2019), we consider this as the baseline model. Table 6 demonstrates the results of this experiment. It can be seen that ETS significantly enhances performance, showing an improvement of approximately 15% in both full fine-tuning and linear probing settings over the baseline model. Meanwhile, adding N3S improves zero-shot classification by 2%, and introducing Flan-T5 enhances performance in linear probing by 4% compared to the baseline. These results underscore the effectiveness of each component in optimizing the model's performance.

**Text Encoders.** In this ablation study, we evaluate the performance of four pre-trained language models, namely Bert (Devlin et al., 2019), Deberta (He et al., 2020), Med-CPT (Jin et al., 2023), and Flan-T5 (Chung et al., 2024) to determine the most suitable language encoder for our model. Here, only the base versions were tested. As shown in Table 7, Flan-T5 outperforms the others across multiple metrics, highlighting the importance of choosing a model that excels not only in general text processing but also in capturing domain-specific nuances, such as ECG reports.

*Table 7.* Effects of different text encoders.

| Text encoder | Full fine-tune | Linear probing | Zero-shot |
|---|---|---|---|
| Flan-T5 | **81.88 ± 3.52** | **80.52 ± 6.08** | **72.50 ± 9.01** |
| Med-CPT | 81.02 ± 3.61 | 79.57 ± 6.32 | 71.81 ± 9.14 |
| Deberta | 79.23 ± 3.65 | 78.24 ± 6.21 | 70.67 ± 9.88 |
| Bert | 78.08 ± 3.91 | 77.58 ± 6.49 | 69.14 ± 9.97 |

**Number of ECG Transformer Layers.** As part of our ablation study, we explore the impact of varying the number of transformer layers (1, 4, 8) in the ECG encoder. As shown in Table 8, increasing the number of layers significantly improves performance. Specifically, the 1-layer model performs 11% worse than the 8-layer model (proposed) in full fine-tuning and 13% worse in linear probing. For zero-shot, the 8-layer model still delivers superior results, with 2% and 3% higher performance than the 4-layer and 1-layer models, respectively. Although these differences are smaller than in full fine-tuning, they still highlight the ECG encoder design's impact on improving performance. It is also worth noting that in this same zero-shot setting (without GPT-4o's support), even our 1-layer case outperformed MERL by approximately 8%, achieving approximately 70% compared to MERL's 62%.

Table 8. Effects of number of transformer layers in ECG encoder.

| # Layers | Full fine-tune | Linear probing | Zero-shot |
|---|---|---|---|
| 8 | $81.88 \pm 3.52$ | $80.52 \pm 6.08$ | $72.50 \pm 9.01$ |
| 4 | $77.63 \pm 4.14$ | $70.17 \pm 7.60$ | $70.64 \pm 8.63$ |
| 1 | $69.40 \pm 4.55$ | $66.83 \pm 7.52$ | $69.43 \pm 9.51$ |

## 5. Conclusion

We propose D-BETA , a novel contrastive masked transformer-based architecture to pre-train ECG signals and corresponding texts. Our approach is generative self-supervised learning, enhanced with ETS loss, and nearest-neighbor negative sampling to support contrastive aspects. Experimental results demonstrate that our method outperforms previous approaches across multiple datasets and on a wide range of downstream tasks with over 100 cardiac conditions. D-BETA shows promise in advancing ECG-based diagnostic models, paving the way for more accurate, efficient, and personalized cardiac care.

## Acknowledgments

This research was supported by the Google South Asia & Southeast Asia research award 2024.

## Impact Statement

Our D-BETA framework advances ECG diagnosis by leveraging multi-modal self-supervised learning, delivering robust benchmarking results across various downstream tasks. We hope D-BETA provides valuable modeling insights for ECG diagnosis and contributes to the healthcare community focused on cardiovascular diseases. We also publicly release the pre-trained models and code, empowering future research and extending ECG analysis to broader clinical applications.

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

# A. Appendix

## A.1. Data and Training Details.

We first visualize representative examples of ECG-text pairs from the MIMIC IV ECG dataset (Gow et al., 2023), as shown in Figure 4. We also indicate the top 30 common unique reports (before merging) in Figure 5. Prominent terms such as "abnormal ecg", "normal ecg", "atrial fibrillation", and "sinus tachycardia" indicate common diagnoses, which suggests prevalent cardiovascular conditions and typical annotations within this dataset.

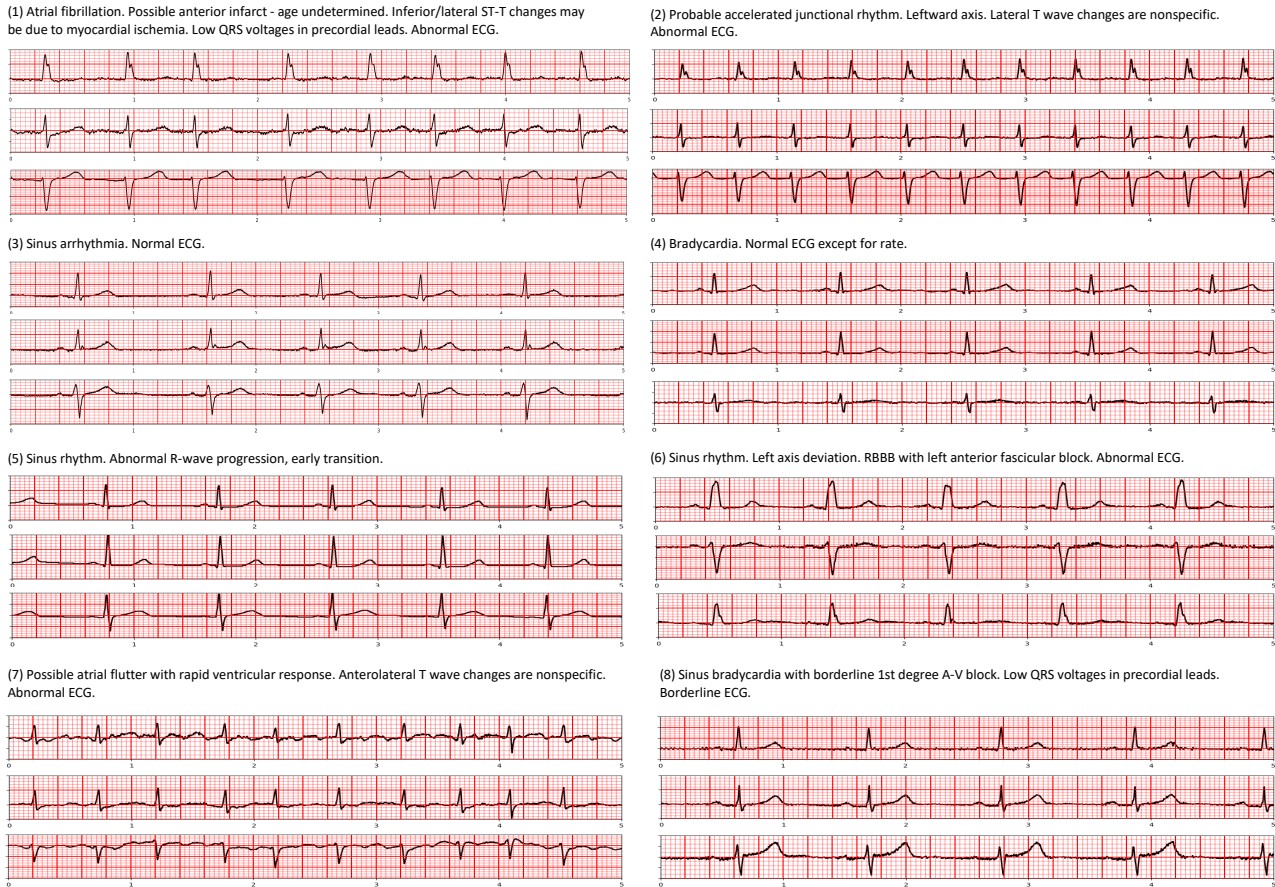

*Figure 4.* Examples of ECG-text pairs in MIMIC IV ECG dataset (Gow et al., 2023). We visualize three leads (I, II, V2) out of twelve.

Next, we provide more details on data configurations in Table 9, including data split, number of classes, metrics, and the corresponding tasks with the given downstream datasets. Particularly, the CODE-test dataset is from the work (Ribeiro et al., 2020) with their model being trained on a set of over 2 million ECG records from 1,676,384 different patients in 811 counties. We show our model's effectiveness by evaluating our performance on the same released test set of 827 samples, but in a zero-shot manner. The samples are originally sampled at 400 Hz, with durations of either 10 seconds or 7 seconds. Therefore, we resampled to 500 Hz and adjusted by truncating or padding with zeros as needed to get 10-second samples. For the gold standard (ground truth), two expert cardiologists provided their diagnoses. If they agree with each other, their consensus becomes the gold standard. In cases of disagreement, a third specialist reviews their diagnoses and determines the final decision.

We also indicate important hyper-parameters during the fine-tuning process in Table 10. We keep training 200 epochs, batch size at 128, and learning rate at 0.001 for the first three datasets. When conducting full fine-tuning experiments, we only need to train 100 epochs and specifically lower the learning rates with 0.00005 and 0.0001 for Dx. and Id. tasks, respectively.

Finally, for the distribution shift experiment, we follow the SCP-codes (classes) matching settings in (Liu et al., 2024b), as shown in Table 11. This is to support three dataset matches (PTBXL-Super and CPSC2018), (PTBXL-Super and CSN), and

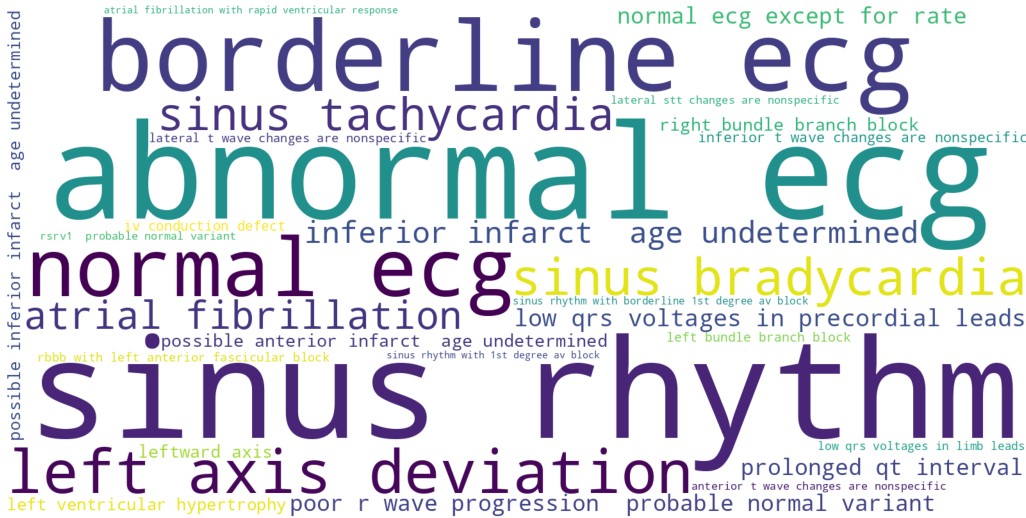

*Figure 5.* WordCloud visualization on the top 30 common unique reports from the MIMIC IV ECG dataset.

*Table 9.* Details on data configurations on five evaluated datasets. Here, LP, ZS are linear probing and zero-shot respectively, while FFT means full fine-tuning.

| Dataset | Tasks | Metric | # Classes | Train | Valid | Test |
|---------|-------|--------|-----------|-------|-------|------|
| PTBXL-Super (Wagner et al., 2020) | LP, ZS | AUC | 5 | 17,084 | 2,146 | 2,158 |
| PTBXL-Sub (Wagner et al., 2020) | LP, ZS | AUC | 23 | 17,084 | 2,146 | 2,158 |
| PTBXL-Form (Wagner et al., 2020) | LP, ZS | AUC | 19 | 7,197 | 901 | 880 |
| PTBXL-Rhythm (Wagner et al., 2020) | LP, ZS | AUC | 12 | 16,832 | 2,100 | 2,098 |
| CPSC2018 (Liu et al., 2018) | LP, ZS | AUC | 9 | 4,950 | 551 | 1,376 |
| CSN (Zheng et al., 2022) | LP, ZS | AUC | 38 | 16,546 | 1,860 | 4,620 |
| Physionet2021-Dx. (Reyna et al., 2021) | FFT | CinC | 26 | 32,640 | 4,079 | 4,079 |
| Physionet2021-Id. (Reyna et al., 2021) | FFT | Accuracy | 2,127 | 147,444 | 17,670 | 2,127 |
| CODE-test (Ribeiro et al., 2020) | ZS | AUC | 6 | – | – | 827 |

(CPSC2018 and CSN). It is worth noting that the None value indicates the target dataset does not have a matching label for given labels in the source dataset.

## A.2. Contrastive Learning Discussion.

**Why Not Use ETM Only in Zero-shot Learning.** As mentioned in the Method section, ETM functions as a contrastive learning technique in the masked auto-encoder architecture. However, it heavily relies on binary classification tasks with explicit ECG-text pairs to learn cross-modal correspondences. It is not designed for zero-shot learning which strongly requires the model to generalize to unseen tasks or classes without the need for such supervised pairings or fused information during training. This motivates us to use ETS , boosting the model's zero-shot ability.

**Why N3S Can Enhance The Performance.** In medical datasets, particularly the MIMIC-IV ECG dataset (Gow et al., 2023), a significant amount of duplicate or highly similar text samples: among nearly 800,000 records, only approximately 180,000 are unique. For instance, over 100,000 samples share an identical text report, which is "sinus rhythm normal ecg". Randomly selecting negative samples for contrastive loss training is not a suitable approach in this scenario. Therefore, we propose using the N3S technique to more effectively differentiate between similar and dissimilar samples, improving contrastive learning by selecting more meaningful negatives. Notably, during training, we observe that the ETM accuracy without N3S stagnates around 75%, while with N3S, it exceeds 96%, demonstrating the significant impact of this approach.

*Table 10.* Details on training configurations on the fine-tuned datasets. For the optimizer, we keep using Adam in all experiments.

| Dataset | # Epoch | Batch size | Learning rate |
|---|---|---|---|
| PTBXL-Super (Wagner et al., 2020) | 200 | 128 | 0.001 |
| PTBXL-Sub (Wagner et al., 2020) | 200 | 128 | 0.001 |
| PTBXL-Form (Wagner et al., 2020) | 200 | 128 | 0.001 |
| PTBXL-Rhythm (Wagner et al., 2020) | 200 | 128 | 0.001 |
| CPSC2018 (Liu et al., 2018) | 200 | 128 | 0.001 |
| CSN (Zheng et al., 2022) | 200 | 128 | 0.001 |
| Physionet2021-Dx. (Reyna et al., 2021) | 100 | 256 | 0.00005 |
| Physionet2021-Id. (Reyna et al., 2021) | 100 | 256 | 0.0001 |

*Table 11.* Domain transfer category matching.

| PTBXL-Super | CPSC2018 |
|---|---|
| HYP | None |
| NORM | NORM |
| CD | 1AVB, CRBBB, CLBBB |
| MI | None |
| STTC | STE, STD |

| PTBXL-Super | CSN |
|---|---|
| HYP | RVH, LVH |
| NORM | SR |
| CD | 2AVB, 2AVB1, 1AVB, AVB, LBBB, RBBB, STDD |
| MI | MI |
| STTC | STTC, STE, TWO, STTU, QTIE, TWC |

| CPSC2018 | CSN |
|---|---|
| AFIB | AFIB |
| VPC | VPB |
| NORM | SR |
| 1AVB | 1AVB |
| CRBBB | RBBB |
| STE | STE |
| PAC | APB |
| CLBBB | LBBB |
| STD | STE, STTC, STTU, STDD |

## A.3. Enhancing Zero-shot Performance with LLMs.

### *(1) Response with merging subtypes reducing capability on new tasks*

"**AFIB**":"**Atrial Fibrillation**, **Paroxysmal** Atrial Fibrillation, **Persistent** Atrial Fibrillation, **Long-standing Persistent** Atrial Fibrillation, **Permanent** Atrial Fibrillation."

"**SEHYP**": "septal hypertrophy, **left** ventricular septal hypertrophy, **right** ventricular septal hypertrophy, apical septal hypertrophy, **mid**-septal hypertrophy."

### *(2) Response showing limitations on LLM's searching and hallucination*

"**AF**": "Atrial Flutter, **Atrial Fibrillation**, Paroxysmal Atrial Flutter, Persistent Atrial Flutter, Long-standing Persistent Atrial Flutter."

"**BIGU**": "**Based on the input, I generated** the following subtypes and attributes for Bigeminal pattern …**Let me know if this meets your requirements!**"

*Figure 6.* Limitations on MERL's enhanced texts.

**Limitations in MERL's Approach.** In zero-shot learning, models typically rely on category names alone to make predictions. However, by incorporating Large Language Models (LLMs), we can enhance the context by generating richer, clinically relevant descriptions of the categories, as discussed in MERL (Liu et al., 2024b). However, we observe two main drawbacks in their enhanced text reports, as shown in Figure 6: 1) MERL's performance heavily depends on their sub-types and attributes searching prompt and additional database. This leads to a limitation when testing detailed analysis with labels that are sub-types themselves. Moreover, this also raises suspicion about the performance when new tasks require labels that are not able to search sub-types and attributes in the database; 2) Following that point, MERL's enhanced texts might be uncontrollable in the outputs where the LLMs provide wrong sub-types or unnecessary context. For example, "Atrial Fibrillation" is already in "AFIB" type but is also shown to be in "AF" with other "Atrial Flutter" types from their settings.

**LLMs in D-BETA .** We address the above points using a straightforward prompt strategy with explicit instructions. Specifically, we employ a prompt: *"You are an experienced cardiologist. For a given clinical term such as 'normal ECG', your job is to describe each term clinically and apply your medical domain knowledge to include other relevant explanations that will help a text encoder like Flan-T5 fully understand medical concepts. Do not include any recommendations in the description."* This makes the LLM generate clinically accurate and more focused explainable descriptions, enhancing the text encoding without introducing irrelevant or redundant information. For example, with the code "AFIB", our prompt on GPT-4o can output: "Atrial Fibrillation (AFIB). Irregular and often rapid heart rate due to uncoordinated atrial activity.".

**Additional Experiments.** Here, we present additional experiments to highlight the effectiveness of ETM loss and masking modeling techniques (e.g., MLM, MEM). Specifically, we perform zero-shot classification with GPT-4o support (reported in AUC (%)) on four datasets: PTBXL-Super, PTBXL-Form, CSN, and CODE-Test.

*Table 12.* Impact of ETM. Results report zero-shot classification in AUC (%).

|         | PTBXL-Super | PTBXL-Form | CSN      | CODE-Test |
|---------|-------------|------------|----------|-----------|
| w/o ETM | 73.2        | 65.8       | **76.6** | 96.2      |
| w ETM   | **76.2**    | **66.1**   | 76.3     | **96.8**  |

As indicated in Table 12, the impact of ETM is demonstrated where, removing ETM slightly decreases performance across most datasets, particularly in PTBXL-Super (76.2 to 73.2). However, the effect on CSN is minimal, suggesting dataset-specific sensitivity to ETM.

*Table 13.* Impact of MLM and MEM. Results report zero-shot classification in AUC (%).

|                 | PTBXL-Super | PTBXL-Form | CSN      | CODE-Test |
|-----------------|-------------|------------|----------|-----------|
| w/o MLM + MEM   | 70.3        | **67.4**   | 74.5     | 94.6      |
| w MLM + MEM     | **76.2**    | 66.1       | **76.3** | **96.8**  |

Finally, we can observe that incorporating MLM and MEM noticeably improves performance across all evaluated datasets in Table 13. Especially, gains are observed in PTBXL-Super (+5.9%), and CODE-Test (+2.2%), demonstrating that the reconstruction tasks also play an important role in enhancing the model's ability for better performance.

