# OpenReview forum: "Boosting Masked ECG-Text Auto-Encoders as Discriminative Learners"
_ICML.cc/2025/Conference — ICML 2025 poster_

### Official Review · Reviewer_TuyF · 2025-03-12

**Overall Recommendation:** 1

**Summary:**

This paper introduces D-BETA, a cross-modal pre-training framework designed for self-supervised learning of ECG signals and textual reports. D-BETA combines the strengths of generative and contrastive learning by leveraging masked language modeling and masked ECG reconstruction to recover missing data. Additionally, it employs contrastive loss functions alongside a nearest-neighbor negative sampling strategy to improve alignment between the two modalities. Extensive experiments conducted on multiple public datasets demonstrate that D-BETA significantly outperforms existing methods, particularly in downstream tasks such as zero-shot learning and linear probing. These findings underscore its strong generalization capabilities and highlight its potential for advancing diagnostic applications.

**Claims And Evidence:**

yes

**Essential References Not Discussed:**

No

**Experimental Designs Or Analyses:**

Yes, the authors implement comprehensive experiments.

**Methods And Evaluation Criteria:**

yes

**Other Comments Or Suggestions:**

please see the weakness.

**Other Strengths And Weaknesses:**

#### **Strengths**
- **Comprehensive Downstream Task Evaluation**: The authors evaluate their model's performance across a wide range of downstream tasks, providing a thorough analysis of its capabilities in practical applications.

#### **Weaknesses**
1. **Lack of Originality in Loss Design**
   - The framework combines multiple loss functions (ETS, ETM, MLM, MEM), none of which are original contributions by the authors. Furthermore, the coefficients used to balance these losses are not systematically explored or justified. This makes the work appear more like an engineering effort rather than a novel machine learning contribution, suggesting it may be better suited for a clinical application journal rather than ICML.

2. **Heavy Reliance on CLIP Loss**
   - In Table 6, removing the ETS loss (which essentially reduces the model without using vanilla CLIP loss) results in a significant drop in performance. This indicates that the model heavily depends on the CLIP loss, while other components contribute relatively little. Similarly, replacing T5 with Med-CPT in Table 7 also leads to a performance decline. These observations raise concerns about whether the improvements stem primarily from leveraging more powerful off-the-shelf text encoders rather than from the proposed framework itself.

3. **Inconsistent Ablation Studies**
   - In Appendix Tables 12 and 13, the authors ablate ETM and MLM losses but fail to provide a consistent and systematic comparison of each individual loss component. This lack of clarity makes it difficult to discern which parts of the framework actually contribute to performance improvements. Additionally, the absence of detailed ablation studies undermines the ability to assess the incremental value of each loss function.

4. **Limited Originality**
   - The work lacks significant novelty, as all the loss functions are borrowed from prior studies, and the authors merely combine them without introducing substantial innovation. This diminishes the originality of the contribution and raises questions about the framework's suitability for a high-impact venue like ICML.

**Questions For Authors:**

please see the weakness.

**Relation To Broader Scientific Literature:**

ECG analysis is a important task for medical application[1].
[1] AI-enabled electrocardiography alert intervention and all-cause mortality: a pragmatic randomized clinical trial. Nature Med

**Theoretical Claims:**

this is not theory work

---

> ### Author Rebuttal · Authors · 2025-03-31
>
> > [R4-1]: Lack of originality  / Limited originality: The framework combines multiple loss functions (ETS, ETM, MLM, MEM), none of which are original contributions by the authors. Furthermore, the coefficients used to balance these losses are not systematically explored or justified. This makes the work appear more like an engineering effort rather than a novel machine learning contribution, suggesting it may be better suited for a clinical application journal rather than ICML … The novelty of the contributions raises questions about the framework's suitability for a high-impact venue like ICML.
>
> We appreciate your feedback and would like to kindly refer to the key contributions of our works as in R2-1. While the individual loss components are inspired by prior works, our contribution lies in how they are specifically adapted, integrated for the underexplored masked ECG-text multimodal autoencoder setting. We acknowledge that we did not systematically explore loss weighting coefficients, which can be a valuable direction for future work. However, **given the context of investigating ECG-text multimodal pre-training by proposing a novel generalized contrastive masked autoencoder framework that has already demonstrated to surpass existing benchmarks, we kindly believe your conclusion about “an engineering effort” only based on the lack loss coefficients may overlook our approach**.
>
> **We also emphasize the growing importance of ECG-based modeling in both machine learning and clinical applications. As you pointed, “ECG analysis is an important task for medical application”, ECGs are non-invasive, widely used physiological signals, and clinical reports provide contextual information that complements the waveform, but the combination between them remains underexplored in dynamic SSLs. Therefore, we hopefully aim for ICML (Primary Area: Applications->Health / Medicine) as a suitable venue to foster this impactful direction.**
>
> > [R4-2]: Heavy reliance on CLIP-style ETS loss and Flan-T5: In Table 6, removing the ETS loss (which essentially reduces the model without using vanilla CLIP loss) results in a significant drop in performance. This indicates that the model heavily depends on the CLIP loss, while other components contribute relatively little. Similarly, replacing T5 with Med-CPT in Table 7 also leads to a performance decline. These observations raise concerns about whether the improvements stem primarily from leveraging more powerful off-the-shelf text encoders rather than from the proposed framework itself.
>
> We would like to respectfully correct a potential misinterpretation in your examples. First, in our attempt to conduct ablation studies, it is ordinary when our proposed components have better performance effects than when not using them. **In Table 6, while removing the ETS loss leads to a performance drop, removing other components (e.g., Flan-T5 or N3S) also results in comparable degradation. In Table 7, Flan-T5 is the encoder we proposed in our framework, and is slightly better than Med-CPT (the best in MERL). We can also see that even with a weaker encoder (e.g., BERT), the remaining architecture still outperforms the last row in Table 6 (same BERT usage)**. Therefore, we sincerely believe the improvements reflect the synergistic design of the overall framework, rather than over-dependence on a specific component.
>
> > [R4-3]: Inconsistent ablation studies on individual losses
> In Appendix Tables 12 and 13, the authors ablate ETM and MLM losses but fail to provide a consistent and systematic comparison of each individual loss component. This lack of clarity makes it difficult to discern which parts of the framework actually contribute to performance improvements. Additionally, the absence of detailed ablation studies undermines the ability to assess the incremental value of each loss function.
>
> **We kindly note that our Table 12 evaluates the effect of ETM, while Table 13 evaluates the joint effect of MLM and MEM purposely, which together form our overall generative reconstruction objective.**
>
> Furthermore, rather than the need of detailed individual loss effect, we believe that focusing more on our main ablation studies which examine the incorporation of core components are more important in our framework, with the effects of N3S, ETS, Flan-T5 (Table 6), variations of text encoders (Table 7), the scalability in the ECG encoder (Table 8).
>
> ---
>
> We hope these clarifications address your concerns. We thank you for your review and hope that you will consider raising our score, as we believe our work offers valuable contributions to the ICML community.

---

### Official Review · Reviewer_ffzJ · 2025-03-12

**Overall Recommendation:** 3

**Summary:**

This paper presents a self-supervised pretraining method for jointly learning from electrocardiograms (ECGs) and text. Their method, D-BETA, combines modality-specific masked modeling, a sigmoid matching loss, and a nearest neighbor negative sampling strategy to enhance performance. Results demonstrate superiority over state-of-the-art ECG pretraining strategies on a variety of downstream datasets and tasks.

## Update after rebuttal
I have read the authors' review and will maintain my original recommendation. I thank the authors for clarifying my questions/concerns, but the rebuttal does not change my overall stance on the submission.

**Claims And Evidence:**

Yes, claims appear to be sound and supported by evidence.

**Essential References Not Discussed:**

C-MELT [1] is another ECG-text pretraining method that combines contrastive and generative objectives; this is an important omission to me.

A few additional citations to ECG-text foundation models could be included [2-4]. I wouldn’t say these are “essential” in that their omission/inclusion changes my stance on the paper, but it is helpful to inform the reader that this is a growing space with many relevant approaches.

[1] C-MELT: Contrastive Enhanced Masked Auto-Encoders for ECG-Language Pre-Training." arXiv preprint arXiv:2410.02131 (2024).

[2] Han, Yu, et al. "Foundation Models in Electrocardiogram: A Review." arXiv preprint arXiv:2410.19877 (2024).

[3] Tian, Yuanyuan, et al. "Foundation model of ECG diagnosis: Diagnostics and explanations of any form and rhythm on ECG." Cell Reports Medicine 5.12 (2024).

[4] Jin, Jiarui, et al. "Reading your heart: Learning ecg words and sentences via pre-training ecg language model." arXiv preprint arXiv:2502.10707 (2025).

**Experimental Designs Or Analyses:**

Experimental design appears sound. I am curious how baseline results were derived, however. Were baselines pretrained from scratch on the same data as D-BETA, were their model weights taken as is and used for fine-tuning, or were results taken directly from their respective papers?

**Methods And Evaluation Criteria:**

Yes. This study uses large-scale, standard ECG datasets and evaluation metrics that are consistent with prior work.

**Other Comments Or Suggestions:**

- L11 on RHS: Change “e.g.” -> “e.g.,”
- L86: Remove extra comma after “D-BETA”
- L374 on RHS: Remove extra space before footnote in “ECG examples ”

**Other Strengths And Weaknesses:**

*Strengths*:
- The paper is generally well-written with clear presentation
- Experiments are thorough and show clear improvement over existing state-of-the-art

*Weaknesses*:
- Some methodological details surrounding baseline implementation could be clarified
- Technical novelty is limited (though the additions are clearly helpful): the sigmoid loss is borrowed from SigLIP – per the authors’ admission – and the combination of contrastive and generative approaches has been seen in C-MELT [1].

**Questions For Authors:**

1.	Were baselines pretrained from scratch on the same data as D-BETA, were their model weights taken as is and used for fine-tuning, or were results taken directly from their respective papers?
2.	What is the authors’ justification for using a language model pretrained on natural text (that is then fine-tuned)? Could the approach potentially be improved by using a text encoder pretrained on cardiovascular reports specifically, perhaps negating the need to fine-tune the text encoder?
3.	Can the authors provide an ablation study on the negative sampling? I see a discussion of why N3S might be helpful in Section A.2, but do not see numerical results backing this up.
4. Do the authors intend to release code and model weights? This will be important to ensure reproducibility.

**Relation To Broader Scientific Literature:**

This study represents an addition to the growing collection of vision-language models for multimodal ECG-text representation learning. This method features a few additional existing previously existing techniques to boost performance beyond current published state-of-the-art.

**Theoretical Claims:**

N/A

---

> ### Author Rebuttal · Authors · 2025-03-31
>
> > [R3-1]: "Were baselines pretrained from scratch on the same data as D-BETA, were their model weights taken as is and used for fine-tuning, or were results taken directly from their respective papers?"
>
> Regarding baseline comparison, we used the results reported in the original baseline papers. They can use **different datasets for the pre-training stage but we keep the fair comparisons by strictly following the baselines’ released data splits, preprocessing and downstream configurations**.
>
> > [R3-2]: "What is the authors’ justification for using a language model pretrained on natural text (that is then fine-tuned)? Could the approach potentially be improved by using a text encoder pretrained on cardiovascular reports specifically, perhaps negating the need to fine-tune the text encoder?"
>
> Thank you for your thoughtful question. **Flan-T5 is pre-trained on large data of natural language that possibly captures rich medical context from domain-specific website (e.g., cardiovascular disease related forums)**, which indicates it is helpful. Additionally, trained with huge corpus, Flan-T5 shows **strong adaptability across unseen tasks. If further fine-tuning Flan-T5 in D-BETA, it could perform better on specific cardiovascular reports**. As shown in Table 7, we can see that Flan-T5 after fine-tuned during D-BETA pre-training stage is nearly 1% better than fine-tuned Med-CPT, a biomedical-pretrained model used in MERL (as a SOTA approach). That said, we agree that using a Flan-T5 text encoder first pre-trained specifically on cardiovascular reports might also offer additional benefits.
>
> > [R3-3]: "Can the authors provide an ablation study on the negative sampling? I see a discussion of why N3S might be helpful in Section A.2, but do not see numerical results backing this up."
>
> We kindly note that the **effectiveness of N3S is already reported in Table 6**, where removing N3S causes a 2% drop in zero-shot performance. Additionally, in Appendix A.2, we report that the ETM accuracy without N3S stagnates at ~75%, while with N3S it exceeds 96%. We believe these results support N3S’s impact of semantically-aware negative sampling.
>
> > [R3-4]: "Do the authors intend to release code and model weights? This will be important to ensure reproducibility."
>
> **Yes, as noted in the Impact Statement part (Lines 442-451), we will publicly release the pretrained models and code upon acceptance. We are happily committed to ensuring reproducibility and enabling future research**.
>
> > [R3-5]: "Technical novelty is limited (though the additions are clearly helpful): the sigmoid loss is borrowed from SigLIP – per the authors’ admission - and the combination of contrastive and generative approaches has been seen in C-MELT [1]."
>
> We respectfully understand the reviewer’s perspective and would like to refer to R2-3 and R2-1 for further discussions on our related justifications and contributions. **We do not directly follow the presentation in their paper but make suitable adjustments in our modeling context and our core contributions go beyond using ETS loss**.  We kindly note that [1] is not discussed here due to the sensitive policy of the conference.
>
> > [R3-6]: “A few additional citations to ECG-text foundation models could be included [2-4]. I wouldn’t say these are “essential” in that their omission/inclusion changes my stance on the paper, but it is helpful to inform the reader that this is a growing space with many relevant approaches.”
>
> Thank you for your kind suggestions. [2] is a comprehensive review paper that synthesizes various aspects of ECG foundation including background, existing datasets, common modeling approaches, and various practical applications. [3] also focuses on the ECG foundation model but proposes to use LLMs to enrich ECG report text with medical knowledge then perform a signal-text-label contrastive learning. Meanwhile, [4] does not pretrain ECG-text directly but interestingly views ECG signals as a language, with QRS complexes as words and rhythms as sentences. **Alongside the discussed works in R1-2, we believe these reflect the growing momentum in this space as you kindly noted**.
>
> ---
>
> We appreciate your time to review our work. At this end, we have addressed your comments and we would be grateful if you are satisfied with our responses and could acknowledge our rebuttal.

---

### Official Review · Reviewer_9VgN · 2025-03-14

**Overall Recommendation:** 2

**Summary:**

This paper introduces the D-BETA framework for joint pre-training of ECG signals and their corresponding clinical text reports, aiming to learn cross-modal self-supervised representations. The method integrates generative tasks—specifically, masked language modeling (MLM) and masked ECG reconstruction (MEM)—with a discriminative task based on ECG-text matching via contrastive learning. Additionally, the paper proposes the ETS loss to enhance the model’s discriminative capability and employs a nearest-neighbor negative sampling strategy (N3S) to effectively select negative samples. Extensive experiments on several public datasets (e.g., PhysioNet 2021, PTB-XL, CSN, CPSC2018, and CODE-test) demonstrate that D-BETA achieves significant performance improvements under full fine-tuning, linear probing (even when only 1% of the training data is used), and zero-shot scenarios. Ablation studies further validate the contributions of key components such as the ETS loss, N3S strategy, and the Flan-T5 text encoder.

**Claims And Evidence:**

Claims:
The authors claim that combining the advantages of generative and discriminative learning can significantly improve the alignment and representation of cross-modal features from ECG signals and text reports, thereby enhancing downstream tasks such as cardiac disease diagnosis and zero-shot inference.

Evidence:
The paper provides extensive experimental results, reporting superior performance (e.g., improvements in AUC, and notable gains in fine-tuning and linear probing scenarios) compared to state-of-the-art methods. In addition, ablation experiments corroborate the positive impact of the ETS loss and the N3S strategy on the overall model performance.

**Essential References Not Discussed:**

Although the paper cites a substantial number of related works, it might benefit from discussing more recent advances in multimodal fusion or self-supervised learning in the context of ECG data to further enrich the literature background and offer a more comprehensive comparison.

**Experimental Designs Or Analyses:**

The experimental design is comprehensive, covering full fine-tuning, low-resource linear probing (using 1% of training data), and zero-shot settings, and validating the model across multiple public ECG datasets. Ablation experiments assess the contributions of critical components (e.g., various text encoders, different numbers of Transformer layers, and the presence or absence of the ETS loss and N3S strategy).

However, it is noted that the experimental results for zero-shot ECG classification are consistent with those reported in “Zero-Shot ECG Classification with Multimodal Learning and Test-time Clinical Knowledge Enhancement” and “Lead-agnostic Self-supervised Learning for Local and Global Representations of Electrocardiogram.” It remains unclear whether the authors have reproduced these works under identical experimental conditions for a fair comparison.

Moreover, the paper does not directly compare the ETS loss against a traditional softmax-based loss; instead, it only shows the effect of removing ETS. This approach demonstrates the contribution of the discriminative task but does not conclusively prove the superiority of the sigmoid-based ETS loss over conventional softmax formulations.

**Methods And Evaluation Criteria:**

Methods:
The paper employs a Transformer-based ECG encoder and a pre-trained Flan-T5 text encoder, using a cross-attention module to fuse features from both modalities. Three task branches are set up corresponding to masked language modeling (MLM), masked ECG reconstruction (MEM), and ECG-text matching (ETM).

Evaluation Criteria:
The experiments are conducted on multiple datasets under full fine-tuning, linear probing, and zero-shot settings, with evaluation metrics mainly including classification accuracy and AUC. Overall, the chosen methods and evaluation criteria effectively capture the characteristics of the problem and the capabilities of the proposed model.

**Other Comments Or Suggestions:**

none

**Other Strengths And Weaknesses:**

Strengths:

- The paper innovatively combines generative and discriminative self-supervised learning.
- The proposed method demonstrates particularly strong performance in low-resource and zero-shot settings, showing promising potential for practical applications.

Weaknesses:

- The overall task design seems to largely build upon existing multimodal joint modeling methods and appears more like an aggregation of several tasks, lacking sufficient novelty.
- Regarding the ETS loss, while the use of a Sigmoid function to avoid the computational burden of global normalization is an interesting idea, the experimental validation and theoretical justification for its effectiveness in enhancing discriminative power remain insufficient.

**Questions For Authors:**

The paper does not include an experiment comparing ETS directly against a conventional softmax-based loss—only an ablation where ETS is removed is provided. How can the authors further demonstrate the efficiency and superiority of the ETS loss over traditional methods?

In the loss module, the authors emphasize the efficiency advantages of the proposed approach. Could the authors elaborate on how the N3S strategy impacts the overall computational burden? Does it affect the model’s efficiency and scalability?

Have the authors reproduced and fairly compared related methods under a unified experimental setting to ensure a fair comparison of performance?

**Relation To Broader Scientific Literature:**

This work is closely related to previous self-supervised ECG learning methods (e.g., CMSC, 3KG, ST-MEM) and multimodal fusion approaches (e.g., MERL), while also drawing inspiration from visual-language pre-training frameworks like CLIP. The innovative incorporation of the N3S strategy using FAISS for efficient vector retrieval and the integration of generative and discriminative tasks (e.g., ETS loss) provide a novel perspective on cross-modal feature learning.

**Theoretical Claims:**

Fusion of Generative and Discriminative Learning:
The authors argue that combining generative tasks (e.g., MLM and MEM) with a discriminative task (ECG-text matching) in a unified self-supervised framework can complement each task’s strengths. While the generative tasks help capture the fine-grained structure of the data, the discriminative task reinforces the separation of cross-modal features, leading to more robust and discriminative representations.

Sigmoid-based ETS Loss:
The paper introduces a novel ETS loss that computes the matching probability for each ECG-text pair independently using the Sigmoid function and optimizes the distance between positive and negative pairs via binary cross-entropy loss. The authors contend that this approach is more efficient than traditional softmax-based contrastive losses—which require global normalization—and thus provides a lightweight, memory-friendly alternative.

Rationale Behind the N3S Strategy:
The authors provide theoretical justification for using the nearest-neighbor negative sampling (N3S) strategy instead of random negative sampling. By selecting negative samples that are semantically distant from the positive samples in the pre-trained text embedding space, the strategy effectively enhances the contrastive learning process. This claim is based on the recognition of inherent semantic similarities in medical text data and underscores the importance of semantic divergence in negative sampling.

---

> ### Author Rebuttal · Authors · 2025-03-31
>
> > [R2-1]: "The overall task design seems to largely build upon existing multimodal joint modeling methods and appears more like an aggregation of several tasks, lacking sufficient novelty."
>
> While our framework builds upon several established tasks, to the best of our knowledge, **we are the first to design and investigate the contrastive masked autoencoder ECG-text pre-training**. As novelty is subjective, we focus on the contribution and significance of our work to the ICML (especially with the primary area of Applications->Health/Medicine) community:
>
>  - We propose D-BETA that specially uses a transformer-based ECG encoder and the Flan-T5 model (overlooked in ECG-clinical community), together with attention-based fusion modules and decoders.
>  - We propose and investigate the insight of discriminative ETS loss into masked ECG-Text autoencoder implementations (with ETM, MLM, MEM and arbitrary lead augmentation), enabling robust multimodal representation learning (unexplored in the literature).
>  - We are the first to introduce the N3S technique to address data unexplored redundancy in the MIMIC-IV ECG dataset, improving the quality of negative samples and boosting model performance.
>  - We conduct extensive experiments across zero-shot, linear probing, and fully fine-tuned settings, demonstrating that our approach consistently outperforms strong baselines across more than 100 cardiac conditions.
>
> We also verify the effectiveness of proposed components by conducting the diverse ablation studies and necessary appendix, as as acknowledged by the reviewers NfLP ("Experimental designs are generally sound"), ffzJ ("Experiments are thorough and show clear improvement over existing state-of-the-art") and TuyF (“providing a thorough analysis of its capabilities in practical applications”). Finally, we emphasize the practical implication of the ECG research field and sincerely believe that our work reflects a principled design and empirical effort rather than simple aggregation.
>
> > [R2-2]: "Although the paper cites a substantial number of related works, it might benefit from discussing more recent advances in multimodal fusion or self-supervised learning in the context of ECG data."
>
> Thank you for the suggestion. This is closely related to R1-2 in which we discuss more recent works regarding multimodal ECG-Text fusion.
>
>
> > [R2-3]: "The paper does not include an experiment comparing ETS directly against a conventional softmax-based loss … can the authors further demonstrate the efficiency and superiority of the ETS loss over traditional methods?"
>
> Thank you for the thoughtful point. Our ETS loss is inspired by the Sigmoid contrastive loss proposed in SigLIP[1], which has already **demonstrated strong theoretical and empirical advantages over softmax-based losses. These “Sigmoid” benefits remain efficiency and superiority in our implementation since we kindly introduce the unaffected ETS heads with Dense layer and Tanh activation while simply setting t=1 and b=0 (as we design positive-negative balanced batches during training)**. Building on this foundation, we focus our main ablations on showing ETS’s benefit in our specific multimodal ECG-text setup.
>
> [1] Zhai, Xiaohua, et al. "Sigmoid loss for language image pre-training." Proceedings of the IEEE/CVF international conference on computer vision. 2023.
>
> > [R2-4]: "In the loss module, the authors emphasize the efficiency advantages of the proposed approach. Could the authors elaborate on how the N3S strategy impacts the overall computational burden? Does it affect the model’s efficiency and scalability?"
>
> We appreciate this point and acknowledge the importance of discussing the computational aspect of using N3S. First, the FAISS index is constructed once before training using precomputed text embeddings in the small Flan-T5 space. During training, we only perform efficient nearest-neighbor retrieval using this index and this action will introduce some tradoff. Importantly, FAISS is known for its impressive speed and scalability in large-scale vector search tasks, and in our case, produces comparable runtime overhead and still doesn’t affect the efficiency and scalability much. **Empirically, on 1× NVIDIA A100-40GB, the one-time loading of the model and FAISS index takes ~1.2 seconds while the average (over 1000 samples) time using FAISS was approximately 0.00219 seconds (compared to 0.00002 seconds without it)**.
>
> > [R2-5]: "Have the authors reproduced and fairly compared related methods under a unified experimental setting to ensure a fair comparison of performance?"
>
> Regarding baseline comparison, we used the results reported in the original baseline papers. In D-BETA, we always aim for the fair comparisons by strictly following the baselines’ released data splits, preprocessing and downstream configurations.
>
> ---
>
> We sincerely appreciate your valuable feedback and have responded to your reviews and hopefully these meet your expectations and acknowledgement.

---

### Official Review · Reviewer_NfLP · 2025-03-15

**Overall Recommendation:** 3

**Summary:**

This paper proposes D-BETA, a novel contrastive masked transformer-based architecture to pre-train ECG signals and corresponding texts. The key components of the proposed approach include self-supervised learning for both ECG and medical texts, as well as fusion mechanism for both to enhance cross-learning. A nearest-neighbor negative sampling was also used to support contrastive learning. Experiment results and ablation studies were included.

**Claims And Evidence:**

Generally well supported claims.

**Essential References Not Discussed:**

NA

**Experimental Designs Or Analyses:**

Experimental designs are generally sound. It would be helpful to also discuss the computational efficiency of the proposed approach, when comparing to similar approaches that can incorporate both ECG and texts.

**Methods And Evaluation Criteria:**

Evaluation criteria are generally sound. One comment/question is that in evaluating and comparing the proposed method to other baselines, whether all other baselines incorporates the text information in training. The text information itself is rich and sometimes contain even more information than the ECG itself, so if some of the other methods do not have the text info available, the comparison may not be a fair one.

**Other Comments Or Suggestions:**

Please explain when applying MAE to the ECG (multi-channel), how are the different channels masked (randomly, same time window masked simultaneously across channels, etc.)

Please explain explicitly the formation of the latent representation $z$ as a fusion of the outputs from the ECG-specific and text-specific encoders. This seems to be the crucial step to enhance cross-learning of the two components.

Overall, the paper would benefit from some theoretical analysis/heuristic on how cross-learning and fusion of the ECG and text component improve model performance.

In some scenarios, the text info, especially from doctors, may be seen as a ground truth or label, rather than training data. Some discussion on this would be helpful, too.

**Other Strengths And Weaknesses:**

None

**Questions For Authors:**

NA

**Relation To Broader Scientific Literature:**

In discussing literature related to using both ECG and text in self-supervised or contrastive learning, the only paper cited was MERL. It would be helpful to expand the literature review here, as this is the key contribution of this paper.

**Theoretical Claims:**

NA

---

> ### Author Rebuttal · Authors · 2025-03-31
>
> We thank the reviewer for their constructive feedback on our submission. We would like to address your comments below:
>
> > [R1-1]: “In some scenarios, the text info, especially from doctors, may be seen as a ground truth or label, rather than training data. Some discussion on this would be helpful, too.”  … “In evaluating and comparing the proposed method to other baselines, whether all other baselines incorporate the text information in training … if some of the other methods do not have the text info available, the comparison may not be a fair one.”
>
> Firstly, our model is not explicitly exposed to ground-truth labels during pre-training. **The textual reports in the MIMIC-IV-ECG dataset are machine-generated and serve as various descriptive inputs, not as discrete predefined label annotations**.
>
> Furthermore, additional text description can be beneficial, but importantly, we presented the efficient way to leverage them in an underexplored multimodal setting, optimizing performance in **the same downstream experiments** with the baselines: (1) Comparing with strong ECG-only SSL methods to show multimodal modeling power. (2) Comparing MERL (ICML publication) in the same multimodal aspect.
>
> Moreover, evaluated **downstream tasks such as fine-tuned classification and identification use unseen ECG recordings, no text is involved; or complete zero-shot tasks** (consistent with the latest close work like MERL). Therefore, we believe these comparisons are both fair and meaningful.
>
>
> > [R1-2]: “It would be helpful to expand the literature review here, as this is the key contribution of this paper.”
>
> **We acknowledge that discussing more details of recent related ECG-text works would be more helpful to our work. Therefore, we kindly synthesize more recent works available, and by this, we also want to highlight the rapidly growing developments and attention to this medical domain**:
>
> Firstly, [1] pretrains on a large private dataset using a standard contrastive SSL with ResNet-like ECG and BERT-based text encoders. Their evaluation is also limited to a few downstream tasks and diseases. [2] build on a similar contrastive modeling design but introduces prompt-based zero-shot inference. While insightful, their evaluation remains relatively narrow. ESI [3] opens another interesting angle when using a RAG pipeline to produce auto-generated detailed report data for the pretraining stage using multiple datasets (including MIMIC IV, PTB-XL, and Chapman). However, the contrastive modeling approach is still relatively similar (Bert-like text encoder, fully convolutional Convnext model, no MEM, ETM, or lead augmentation), and evaluation is relatively modest.
>
> Recently, as ECG representation learning continues to evolve, a few works have begun to explore more practical, user-end applications. For example, [4,5] leverage instructed multimodal LLMs to deal with ECG report generation and clinical question answering. These methods largely benefit from a well-pretrained ECG encoder, which D-BETA also can be potentially adapted.
>
> [1] Lalam, Sravan Kumar, et al. "Ecg representation learning with multi-modal ehr data." TMLR (2023).
>
> [2] Liu, Che, et al. "Etp: Learning transferable ecg representations via ecg-text pre-training." ICASSP (2024).
>
> [3] Yu, Han , et al. "Ecg semantic integrator (esi): A foundation ecg model pretrained with llm-enhanced cardiological text." (2024).
>
> [4] Zhao, Yubao, et al. "ECG-Chat: A Large ECG-Language Model for Cardiac Disease Diagnosis." (2024).
>
> [5] Yang, Kai, et al. "ECG-LM: Understanding Electrocardiogram with a Large Language Model." HDS (2025).
>
>
> > [R1-3]: “Please explain when applying MAE to the ECG (multi-channel), how are the different channels masked”
>
> As described in Section 3.1 (Lines 139-143), we apply random lead masking, where entire ECG channels are independently masked with a probability of 0.5. This lead-wise masking encourages the model to learn robust representations across varying input configurations.
>
>
> > [R1-4]: “Please explain explicitly the formation of the latent representation z as a fusion of the outputs from the ECG-specific and text-specific encoders. Overall, the paper would benefit from some theoretical analysis/heuristic on how cross-learning and fusion improve model performance.”
>
> Our fusion module is designed as stacked cross-attention blocks, which allows interaction between multi-lead ECG signals and semantic text embeddings (Lines 175-180) before decoding. This produces essential joint embeddings for our learning objectives: for example, the ETM task requires a unified representation to determine whether an ECG-text pair is matched, which naturally depends on information from both modalities. Similarly, MLM and MEM benefit as one modality provides useful context to reconstruct the other (e.g., ECG noise or text ambiguity compensation). This fusion also facilitates future downstream tasks such as ECG report generation, where signal-based text generation is critical.

---

### Decision · Program_Chairs · 2025-05-01

**Decision:**

Accept (poster)

**Comment:**

We thank the authors for their submission.  The study is interesting, well written, and contains a thorough empirical evaluation across a variety of ECG datasets and tasks.  Reviewers shared the concern that the technical approach contains limited novelty.  However, the main contribution is focused on the applied ECG domain, and though there is limited novel ML, the succinctly presented and thorough ablation study conducted show that this set of components appears to meaningful imbue ECG representations with valuable information.  As reviewer ffzJ puts it, "Experiments are thorough and show clear improvement over existing state-of-the-art."  During the rebuttal period, authors clarified points to valid questions raised in the initial round of reviews.  The authors are encouraged to incorporate those clarifying points into their manuscript.